# Reducing off-Flavour in Commercially Available Polyhydroxyalkanoate Materials by Autooxidation through Compounding with Organoclays

**DOI:** 10.3390/polym11060945

**Published:** 2019-05-31

**Authors:** Lidia García-Quiles, Arantzazu Valdés, Ángel Fernández Cuello, Alfonso Jiménez, María del Carmen Garrigós, Pere Castell

**Affiliations:** 1Tecnopackaging, Polígono Industrial Empresarium C/Romero N° 12, 50720 Zaragoza, Spain; 2Analytical Chemistry, Nutrition & Food Sciences Department, University of Alicante, P.O. Box 99, 03080 Alicante, Spain; arancha.valdes@ua.es; 3Escuela de Ingeniería y Arquitectura, University of Zaragoza, Av. María de Luna, 3, 50018 Zaragoza, Spain; afernan@unizar.es; 4NANOBIOPOL Research Group, University of Alicante, San Vicente del Raspeig, 03690 Alicante, Spain; alfjimenez@ua.es (A.J.); mc.garrigos@ua.es (M.d.C.G.); 5Fundación Aitiip, Polígono Industrial Empresarium C/Romero N° 12, 50720 Zaragoza, Spain

**Keywords:** biopolymers, nanoclays, bio-nanocomposites, extrusion-compounding, polyhydroxyalkanoates, thermal properties, microstructure, volatiles, autoxidation, thermal gravimetric analysis, scanning electron microscope, headspace solid phase microextraction

## Abstract

Polyhydroxyalkanoates (PHAs) are nowadays considered competent candidates to replace traditional plastics in several market sectors. However, commercial PHA materials exhibit unsatisfactory smells that can negatively affect the quality of the final product. The cause of this typical rancid odour is attributed to oxidized cell membrane glycolipids, coming from Gram-negative production strains, which remain frequently attached to PHAs granules after the extraction stage. The aim of this research is the development of customised PHA bio-nano-composites for industrial applications containing organomodified nanoclays with high adsorbance properties able to capture volatile compounds responsible for the displeasing fragrance. To this end, a methodology for the detection and identification of the key volatiles released due to oxidative degradation of PHAs has been established using a headspace solid-phase microextraction technique. We report the development of nine bio-nano-composite materials based on three types of commercial PHA matrices loaded with three species of nanoclays which represent a different polar behaviour. It has been demonstrated that although the reached outcoming effect depends on the volatile nature, natural sepiolite might result in the most versatile candidate for any the PHA matrices selected.

## 1. Introduction

The poly(hydroxyalkanoates) (PHAs) market is poised to grow during the next decade at a compounded annual growth rate of 6.3% gaining more and more attention in the biopolymers market [1]. PHA biopolyesters are increasingly used owing to its biodegradable nature and processing versatility representing a potential sustainable replacement for fossil oil-based commodities [2]. PHA biopolymers are formed mainly from saturated and unsaturated hydroxyalkanoic acids in which the monomer unit harbours a side chain R group which is usually a saturated alkyl group. These polymers are generally classified in two categories depending on the number of carbon atoms in their monomer units: small chain length (*scl*)-PHA when the monomer units contain from three to five carbon atoms and medium chain length (mcl)-PHA with monomer units possessing from six to 14 carbon atoms [3,4]. These features give rise to diverse PHA monomers and polymers with tailored molecular weights and melting points providing broad properties, such as isotactic poly(3-hydroxybutyrate) P3HB; poly(3-hydroxybutyrate-*co*-hydroxyvalerate) PHBV copolymer; 3-hydroxypropionate (3HP); 3-hydroxyoctanoate (3HO); or 4-hydroxybutyrate (4HB) among others [2,5].

Accumulation of PHA seems to be a common metabolic strategy adopted by many bacteria to cope with a series of various stress factors in the environment [6]. PHA biopolyesters are used as carbon and energy storage and are accumulated by various prokaryotes as intracellular “carbonosomes” [7] in the form of granules of 0.2–0.5 microns inside these bacterial cells in an amorphous state (chain disordered) are covered by an outer monolayer of phospholipids and proteins [8]. To recover the PHA granules is necessary to break bacterial cells, remove the monolayer and isolate the PHA with high molecular chains (unbroken chains). To this end, a combination of solvents is used, usually based in organic solvents and/or chlorinated. These solvents modify the cell membrane permeability and dissolve the PHA, which is able to escape through the monolayer and cell wall [9,10]. Recent works have studied the effect that carbon sources may have on the final odours released by PHA monomers [11].

The proposal of efficient methods to purify microbial PHAs to meet the legislative requirements in the pharmaceutical, medical or food sectors has been investigated [12,13]. Simple chemical methods, such as combinations of organic PHA solvents and anti-solvent or high-pressure extraction, can be applied to remove the remaining lipids and endotoxins that are frequently attached to PHAs from Gram-negative production strains [14]. In the case of PHAs, lipid residues often remain attached to the biopolymer after extraction, causing a typical rancid odour and smell that can negatively affect the quality of the final product. The flavour threshold of a particular chemical can vary greatly and is defined as the minimum quantity of a substance, which can be detected by 50% of the taste panel [15]. Regardless of the oxidation mechanism, it is recognised that lipids are oxidised to odourless and flavourless intermediates that could break into molecules giving off-flavours. These conditions are associated with the production of free radicals by autoxidation which have been recognized as a potential source of food quality shortcomings of PHA for many applications such as packaging [16,17]. Nowadays, plastic product converters purchase commercial PHA materials with unsatisfactory smelling because current PHA production methods are not solving this particular disadvantage. Our proposed solution approaches the compounding stage of customised blends for industrial applications by using nanoclays with high adsorbance properties able to capture volatile compounds responsible for the displeasing fragrance.

Up to now, microextraction methods have been used for the separation of specific analytes from complex matrices with high reproducibility, selectivity and sensitivity [18]. This is a very important process commonly used in the food sector in order to analyse the aroma properties. Among several extraction methods widely used for the extraction and determination of flavour compounds, the most frequently applied are those based on headspace (HS) analysis. Solid phase microextraction (SPME) is a sample preparation technique based on sorption that constitutes a reliable tool for the analysis of organic volatile and semi-volatile compounds [19]. In the plastic sector, the analysis of volatile organic compounds emissions from plastic and rubber materials through HS-SPME and other techniques such as (MAE)-GC-MS, (SPM)-GC-MS, (TGA)-GC-MS etc have been performed as a useful method for understanding polymer degradation and object damage made of conventional oil-based polymers such as high-density polyethylene (HDPE), low density polyethylene (LDPE), polyacrylonitrile (PAN), polyvinyl chloride (PVC), polypropylene (PP), polystyrene (PS), polymethyl methacrylate (PMMA), polyurethane (PU), and nylon 6,6 (PA) [20,21,22] Specific volatile emissions can be related to deterioration, including loss of additives and polymer degradation [23].

However, headspace solid-phase microextraction (HS-SPME) has not been used for the analysis of those volatile compounds released from PHAs. This technique used by the authors is combined with gas chromatography ion-trap/mass spectrometry (GC-IT/MS) in order to quantify a large number of volatile compounds in PHA samples. To the best of our knowledge, this is the first time that a method for analysis of volatiles is used in PHA final/commercial materials for such purpose. 

In the present work, three PHA materials were reinforced with organically modified and unmodified Sepiolite and Montmorillonite as described in the materials section. Clays and zeolite exchanged with various cationic surfactants have been shown to be effective adsorbents for a variety of organic compounds [24]. Sepiolite has outstanding sorption capacity and may adsorb a large variety of molecules as vapours as well as liquids due to its natural structure that forms inner zeolitic channels that provides a high microporosity and large specific surface area [25]. Sepiolite fibre leads to the presence of numerous silanol groups (Si–OH) all along the edges of the fibre [26] which are useful as fillers providing reinforcing characteristics with polar polymers [27]. Sodium montmorillonite (Na-MMT) is a smectite nanoclay that has two tetrahedral sheets of silica sandwiching a central octahedral sheet of alumina and largely used in industry too due to its swelling and adsorption properties [28,29].

The aim of this research was the development and characterization of novel composite materials based on different PHA matrices to diminish the release of undesirable oxidised lipid molecules through the introduction of organoclays. The objective is to establish a methodology for the identification of the key volatiles released by commercial PHAs which are responsible for unpleasant odours, and demonstrate that the introduction of commercial organoclays can be an efficient approach to capture those volatiles and reduce the displeasing aroma. The studied techniques used in this paper can be useful for processors in order to monitor the oxidative degradation of PHAs with storage time and to evaluate their acceptance on the market.

## 2. Materials and Methods 

### 2.1. Materials

Three commercial PHA materials were used as a matrix. These were Mirel PHA1005 and Mirel PHA3002 (food contact P3HB-*co*-P4HB, purchased from Metabolix, Cambridge, MA, USA), and Biomer PHB P226 (homopolymer P3HB grade purchased from Biomer, Krailling, Germany).

Three different modified and unmodified organoclays were kindly provided by TOLSA (Madrid, Spain). A natural sepiolite (T2), a modified Sepiolite with aminosilane groups on its surface (T1) and a sodium montmorillonite modified with a quaternary ammonium salt (T3). The three candidates selected present different behaviours from polar (T1) and neutral (T2) to non-polar (T3) feature which directly affects their affinity for the PHAs polyesters (Table 1).

Chemical patterns for volatile quantification (considered as main responsible for the odour), were purchased from Sigma-Aldrich. These are 1-hexanol, heptanal, octanal, decanal, α-Methylstyrene, 4-methylstyrene and Benzaldehyde.

### 2.2. Nano-Bio-Composites Preparation

A 26-mm twin-screw Coperion ZSK 26 compounder machine (Stuttgart, Germany) was used to prepare PHA/nanoclay formulations by extrusion-compounding. Twelve different formulations were studied in total accounting for the three control matrixes plus nine composite materials loading them with the three nanoclays at 3 wt % in all cases. The melted polymers and nanoclay powder were mixed at a screw speed of 125 rpm; temperature was increased from 150 °C in the feeding zone up to 165 °C at the nozzle for PHA1005 and PHA3002 (P3HB-*co*-P4HB formulations) and slightly decreased from 140 °C in the feeding zone up to 160 °C at the nozzle when blending PHB226 (P3HB). The compound was extruded through a 2 mm diameter die for a constant output of 10 kg/h. The extrudate was quenched in a water bath at room temperature, dried and cut into pellets. A total amount of 3 to 5 kg per material was produced.

Specimens for mechanical testing were obtained by injection moulding with a JSW 85 EL II electric injection machine (Tokyo, Japan) by following ISO 178 and ISO 527 standards. These were tested and broken samples were used to study the structural properties (results on mechanical properties are out of the scope of this paper). A complete characterization covering the structural, thermal and mechanical behaviour of the formulations developed, understanding the compatibility mechanisms between the different organoclays and the matrices can be found in a previous work carried out by the authors and published at García-Quiles et al.) [30].

### 2.3. General Characterisation Methods

#### 2.3.1. Scanning Electron Microscopy

Structural properties were evaluated by scanning electron microscopy (SEM) with Hitachi S3400N equipment (Tokyo, Japan) in order to determine the morphology and dispersion of the nanoclay through the biopolymer-based materials. Samples for SEM observation were obtained from fractured mechanical specimens. 

Energy-dispersive X-ray spectroscopy (EDX) was also used for samples for chemical characterization. 

#### 2.3.2. Thermogravimetric Analysis

Thermal characterisation was carried out by using thermogravimetric analysis (Mettler-Toledo TGA/SDTA 851e, Greifensee, Switzerland). Samples (5.0 ± 0.1 mg) were heated from 25 to 800 °C at 5 °C/min under N_2_ atmosphere (50 mL/min).

#### 2.3.3. Analysis of Volatile Compounds by HS-SPME-GC-MS

##### HS-SPME Extraction Procedure

The sample (1.00 ± 0.01 g) was mixed with 1 mL of distilled water and 2 μL of internal standard 4-methylstyrene (8 mg/kg, Sigma-Aldrich Inc., St. Louis, MO) and a small magnetic stirrer were placed in a 20 mL amber vial sealed with an aluminium pressure cap provided with a pierceable silicone septum. The extraction was carried out using the SPME fibre made of divinylbenzene/carboxen/polydimethylsiloxane (DVB/CAR/PDMS) 50/30 mm, StableFlex, 1 cm long, mounted on a manual support set SPME (Supelco, Bellefonte, PA, USA) [31,32]. The vial with the sample was placed in a water bath under a specific temperature. After 10 min of equilibrium of the sample, the SPME needle was inserted into the vial and exposed to the headspace of the vial for a specific time. After extraction, the fibre desorption was performed in the GC-MS injection port at 250 °C for 12 min (splitless mode). Blank tests were performed prior to the analysis of the samples to ensure that there was no contamination that could cause memory effects.

##### GC-MS Parameters

The analysis of the volatiles was carried out using an Agilent 6890N GC model gas chromatograph coupled to an Agilent 5973N MS mass spectrometer (Agilent Technologies, Palo Alto, CA, USA) with impact ionization source of electrons (EI 70 eV) and quadrupole analyzer. The temperatures of the transfer line “ion source” and GC-MS were 230 and 250 °C, respectively. A TRB-624 column (30 m × 0.25 mm × 1.4 μm) of Teknokroma was used, which was programmed from 50 °C (maintaining 2 min) to 250 °C (isotherm for 7 min) at 10 °C/min. Helium was used as the carrier gas (1.5 mL/min). The volatile compounds were identified in “scan” mode (m/z 30–550) by comparing their mass spectra with those of the standard compounds and the Wiley compound library. When standards were not available, volatile compounds having ≥80% similarity with Wiley library were tentatively identified using GC-MS spectra only. All patterns obtained from the commercial house Sigma-Aldrich, St. Louis, were quantified by HS-SPME-GC-MS using standard calibration curves. For this, mother solution (5 mg/kg) and working solutions were prepared using distilled water as a solvent. The HS-SPME and GC-MS sampling procedures were carried out as described for the samples. The analytical method used for the quantification was validated in terms of linearity, repeatability and limits of detection (LOD) and quantification (LOQ). The calibration curves were performed at five concentration levels, in triplicate, using adequately diluted standards and adjusted by linear regression.

##### Optimization of HS-SPME Extraction Parameters Box-Benhken Experimental Design (BBD)

The extraction of volatile compounds from PHAs was performed under different extraction conditions according to the experimental design shown in Table 2. The parameters considered during HS-SPME optimization were extraction temperature (50, 70 and 90 °C), and extraction time (15, 37.5 and 60 min) and sodium chloride addition (0, 0.5 and 1 M) in a final vial volume of 1 mL. The P3HB sample was selected for the optimization of HS-SPME conditions. The range of studied variables was selected on the basis of results obtained in previous studies reported in the literature [31,32,33]. A Box–Behnken design (BBD), comprising 12 experimental runs and four central points, was used, and experiments were carried out in randomized order from the experimental design were evaluated in terms of the sum of the peak areas of volatile compounds identified with a library similarity higher than 80%

#### 2.3.4. Statistical Analysis

Statgraphics-Plus software 5.1 (Statistical Graphics, Rockville, MD, USA) was employed to generate and analyse the results of the BBD. Statistical significance of model parameters was determined at the 5% probability level (α = 0.05).

All analytical tests were performed in triplicate. ANalysis Of VAriance (ANOVA) was applied to the results by using SPSS software (Version 15.0, Chicago, IL, USA). Tukey’s test was used to assess differences between means and the significance of differences was considered at the level of *p* < 0.05.

## 3. Results & Discussion

### 3.1. SEM—Scanning Electron Microscopy

SEM micrographs (Figure 1) showed the dispersion and affinity between the nanoclays and each of the biopolymer matrices used. The good dispersion, debundling (T1 and T2) and exfoliation (T3) of the nanoclays may highly increase the capture of volatiles. The natural chemical behaviour (higher or lower polarity) of each organoclay surface show a clear effect in the interaction mode with each biopolymer matrix. The organophilic clays are not compatible with hydrophobic organic matrices as the spacing between the nanoclay sheets (T3) and inner channels (T1 and T2) is extremely narrow, and hence diffusion of polymer chains in the nanoclay galleries is not possible. This often leads to aggregation of clay particles, acting as stress-concentration sites in the polymer matrix [34]. The organic modification aims at broadening these channels and helps the polymer to penetrate and get fixed onto the surfaces.

On the one hand, analysing PHA3002 micrographs, T2 seems to appear better dispersed than T1 or at least better oriented in one direction. In addition, sepiolites (T1 and T2) appear to show a better affinity with this P3HB-co-P4HB matrix as T3 micrographs show a lack of wetting contact layer or interphase between the nanoclay and the polymer. On the other hand, for PHA1005 matrix, T1 presents some agglomerates, which in principle should be avoided due to the aminosilane modification. At times, the greater the amount of organic modifier in nanoclays, the greater the impediment to debundle [35] and this might be the behaviour observed between T1 (modified) and T2 (natural) for this matrix.

Finally, when analysing PHB226 compounds, a clear lack of affinity for T2 PHB226 was observed. Similar behaviour was found for T1 in which a similar bubble-like reaction during the blend compounding might have started. However, the polar affinity between T1 and PHB226 probably cuts the kinetics of the reaction while for T2 (non-modified) the process is more pronounced. Therefore, a better interphase interaction between T1 and PHB226 can be observed when comparing to T2. This enhanced interaction was sought with the organoclay modification. However, a completely different affinity behaviour can be found for T3 and PHB226, for which the surface structure complies with a more ductile fracture. Montmorillonite layers are found well integrated and interphases are not discernible as in the other two cases. These results are in agreement with Mechanical and XRD results already published by authors, in which Montmorillonite seemed to be highly intercalated and even exfoliated, while sepiolite debundling was not so effective [30]. This favourable interaction between T3 and PHB226 may be attributed to the lack of P4HB and the appearance of PBA, which modifies the polar behaviour or the matrix, and indirectly affects the overall viscosity of the molten polymer when the blend is developed favouring or hindering miscibility [36].

Finally, for neat matrices, the difference in the amount of talc used by the manufacturer for PHA1005 and PHA3002 formulations respect to PHB226 can be observed, which seems in coherence with TGA findings.

### 3.2. TGA—Thermogravimetric Analysis

The thermal stability of the nanocomposite materials was evaluated through TGA. The temperature corresponding to the onset of decomposition (T onset) for the samples studied is essential for evaluating their thermal stability and is shown in Table 3.

In addition, the appearance of volatile traces in the first section of the DTG was evaluated. Theoretically, high amounts of some identified volatiles that might be found in the neat matrices, may have been released during the extrusion-compounding of the bio-nanocomposites due to the temperature profile reached during the material development (over 150 °C) [21,32]. However, the release rate is unknown and there is not a clear perception of volatile release. Nevertheless, the DTG range between 100 and 220 °C was studied trying to observe substantial peaks. The signal noise of the apparatus seems too high compared to the amount of volatiles released in order to be able to obtain reliable results. Therefore the correct volatile analysis is left for the accurate technique HS-SPME and authors will focus the TGA results to study the thermal stability and the non-organic residue.

The degradation of PHA 3002 occurred in one step while PHA 1005F shows a smooth second step at higher temperatures which may correspond to a greater amount of inorganic fillers. Several steps occurred for the PHB226 sample too, distinctly exhibiting two decomposition platforms. This behaviour has been previously reported in the literature for some pure P3HB polymers and for different P3HB-*co*-P4HB co-blends. According to literature, pure P3HB thermally decomposes at around 270 °C, above its melting point (around 180 °C). However, a short exposure of PHB to temperatures near its melting point induce a severe degradation producing degraded products such as olefinic and carboxylic acid compounds due to the random chain scission reaction that takes place [37]. Wang et al. studied the thermal degradation of P3HB-*co*-P4HB composites [38]. The results showed that thermal degradation of P3HB occurs almost exclusively by random chain-scission involving a six-member ring formation while P4HB decomposes through intermolecular ester exchange reaction and generates a decomposition product ɣ-butyrolactone. Therefore, P4HB crystals are thermally more stable than P3HB. In addition, in a previous analysis carried out by the authors for these composites, the presence of PBA was corroborated in the NMR and DSC analyses for PHB226 [30]. Maximum degradation temperature of PBA is defined at T_dmax_ = 385.1 °C [39] which is aligned with the TGA results shown in Table 3. The presence of multi-component nature for PHB226 commercial formulation containing organic additives and polyesters has been also corroborated by further authors such as Y.M. Corre et al. [40]. 

Different residual weights remaining at 800 °C can be found for the three matrices being higher for P3HB-co-P4HB blends than for P3HB. Usually, two inorganic fillers are commonly used in polymers as nucleating agents to reduce crystallization rates: Mg_3_Si_4_O_10_(OH)_2_ and CaCO_3_ [41,42]. Complementary EDS was used for the elemental analysis of the final residues revealing their correspondence to talc. In addition, the final amount of inorganic residue found for the bio-nano-composites developed, reveals an effective dosage of the nanoclays load introduced.

The thermal degradation of Sepiolite and montmorillonite nanoclay additives was characterised in order to better understand the effect these have over the final bio-nano-composites. 

The T2 thermograph (Figure 2a) shows a typical curve for non-modified sepiolite. T1 thermograph shows a few deviations from respect to T2 due to the modification through organosilanes containing amine groups [25,27]. Our TGA results are in concordance with the aminosilane grafted Sepiolite (T1) characterisation made by G. Tartaglione et al., in previous publication [43]. In particular, four degradation steps are appreciated for T1. The clay presents degradations around 74, 232 and 305 °C. In addition, a less defined degradation phase is observed at 349–495 °C. To try to better define this area and separate the overlap of peaks, the heating rate was lowered from 10 to 5 °C/min. The first loss corresponds to mainly methanol, probably used as a solvent for the modification of the clay, followed by dehydration of water: moisture and zeolitic water (below 180 °C). 

The volatilization of the modifier takes place in two steps, the first one related to adsorbed molecules (non-grafted) at 215–250 °C. The grafted modifier is likely to volatilize over 400 °C. The grafted groups are very stable, being eliminated by heating at temperatures above 400 °C. In this way, the characteristic hydrophilic surface of Sepiolite becomes organophilic and the fibrous clays can then be easily dispersed in low-polar polymers [44]. The last loss observed approximately at 750 °C has been described in the literature to be related to a complete oxidation of the carbonaceous residue formed during previous thermal oxidation of the grafted molecules [45].

T3 (Na-MMT) presents a chemical structure modified with a quaternary alkyl ammonium surfactant which has a noticeable effect on the thermal stability of the organoclay itself. Analysing its thermal degradation behaviour, it can be observed that T3 DTG shows a water/solvent release at 68 °C and at 116 °C. In addition, two small mass loss peaks are found for 295 °C and for 526 °C, and finally an accentuated peak corresponding to 674 °C. According to the literature (Botana et al. [46] and Cervantes et al. [47]) free water loss in MMT nanoclays containing hydroxyl groups in the alkylammonium anion, shows free water loss at around 80 °C. Peaks in at 200–600 °C are attributed not only to structural water but also to the decomposition of the alkylammonium ions. In particular, peaks found at 297 and 528 °C are associated to the following chemical species for this organoclay sample: H_2_O, CO_2_, alkanes, alkenes, CHO’s, COOH’s, amines [47]. Finally, peaks attributed between 610 and 674 °C are attributed to some further structural water release.

In all the developed bio-nano-composites a decrease in T_onset_ and T_max_ can be appreciated except for PHB226-T1. Han et al. reported that P3HB-*co*-P4HB samples containing a 10 wt % silica presented a higher T_onset_ and T_max_ than that of pure P3HB-*co*-P4HB, in particular being 2.08 and 6.28 °C higher, respectively [48]. Our samples containing a 3 wt % of nanoclay content seems not to induce a high thermal stability increase. Nevertheless, results show that sepiolites (T1 and T2) seem more suitable for these polyhydroxyalkanoates materials than montmorillonite for thermal stability purposes, as sepiolites maintain the thermal stability of the original matrix. For P3HB-*co*-P4HB composites, it can be observed that T1 presents the same T_onset_, T_max_ and T_50_ while a slight decrease is found for T2. For T3, the overall decrease for T_onset_, T_max_ and T_50_ is noticeable. A similar tendency is found for P3HB composites. T1 is the nanoclay inducing a higher thermal stability for both degradation steps (including PBA), while T2 tends to maintain the original matrix behaviour and T3 induces a remarkable decrease. Results are detailed in Table 3.

### 3.3. HS-SPME-GC-MS—Headspace Solid-Phase Microextraction Coupled to Gas Chromatography-Mass Spectrometry

#### 3.3.1. Optimization of the HS-SPME Extraction Process

Seventeen different volatile compounds were identified with a library similarity higher than 80% in all runs of the BBD carried out in this study (Figure 3): 1, 1-butanol; 2, p-xylan; 3, heptanal; 4, α-methylstyrene; 5, benzaldehyde; 6, octanal; 7, limonene; 8, 1-hexanol; 9, undecane; 10, 1-octanol; 11, nonanal; 12, decanal; 13, 1-chloro-decane; 14, 1-decanol; 15, tetradecane; 16, biphenyl; 17, 2,6-bis (1-methylethyl)-benzeneamide.

Benzaldehyde, heptanal, octanal, nonanal and decanal belong to the family of aldehydes. The presence of benzaldehyde in other matrices such as polystyrene has been related to oxidation processes together with p-xylan and α-methylstyrene [21]. Furthermore, heptanal, octanal, nonanal, decanal, are secondary oxidation compounds, which are characterized by strong and unpleasant odours characteristic of lipid degradation [49]. In addition, four alcohols were identified (butanol, 1-hexanol, octanol and decanol). In particular, 1-hexanol is one of the most important alcohols used for several processes of synthesis and degradation product of one of the most used plasticizers since 1950, the phthalate DEHP [50]. This compound has been detected in emissions of different plastics with undesirable odour [51]. As expected, some organic volatile compounds identified are linear or branched chain alkanes such as undecane and tetradecane due to the high affinity for SPME fibre coating [32]. Regarding limonene, it is a terpene which has acquired great importance in recent years due to its demand as a biodegradable solvent. Apart from industrial solvent it also has applications as an aromatic component and is widely used to synthesize new compounds [52].

The influence of extraction temperature (A), extraction time (B) and NaCl addition (C) in the sum of the peak areas of these seventeen compounds was evaluated by using standardized Pareto diagrams (Figure 4). This kind of diagram allows for the determination of the magnitude and the importance of each independent variable or factors in a response. Figure 4 shows that only the extraction time was significant and it has a positive effect. That means that extraction by HS-SPME improves when the time increases from 15 min to 60 min. Nonetheless, the extraction temperature and the addition of NaCl and all possible interactions (AB, AC, BC, AA, BB, CC) do not have a significant effect with a confidence level of 95% (α = 0.05). Based on the obtained results, the optimum HS-SPME extraction conditions were 60 min at 90 °C without the addition of NaCl. 

Finally, it was decided to quantified six main volatile compounds in the present work as a result of this negative effect on the final odour of samples: heptanal, α-methylstyrene, benzaldehyde, octanal, 1-hexanol and decanal. The adequacy of the fitted models was determined by evaluating the lack of fit, the coefficient of determination (R^2^) and F test obtained from the analysis of variance (ANOVA). The computing program showed that the fitted models were considered satisfactory as the lack of fit was not significant with values of 0.852 for heptanal, 0.659 for α-methylstyrene, 0.457 for benzaldehyde, 0.940 for octanal, 0.816 for 1-hexanol and 0.983 for decanal. (*p* > 0.05). On the other hand, R^2^ is defined as the ratio of the explained variation to the total variation and is a measurement of the degree of fitness. The model can fit well with the actual data when R^2^ approaches unity with values of 0.733 for heptanal, 0.767 for α-methylstyrene, 0.775 for benzaldehyde, 0.861 for octanal, 0.853 for 1-hexanol and 0.681 for decanal. These values indicated a relatively high degree of correlation between the actual data and predicted values, indicating that models could be used to predict the studied responses.

#### 3.3.2. Validation method

The analytical method used for the quantification of volatile compounds by HS-SPME-GC-MS was validated in terms of linearity, repeatability and detection (LOD) and quantitation (LOQ) limits. An acceptable level of linearity was obtained for all analytes (R^2^ between 0.9076 and 0.9929), showing relative standard deviation (RSD) values lower than 5%. LOD and LOQ values were determined using the regression parameters of the calibration curves (3 Sy/x/a and 10 Sy/x/a, respectively, where Sy/x is the standard deviation of the residues and “a” is the curve slope). LOD and LOQ values obtained for heptanal ranged between 0.001 and 0.004 μg/kg, α-methylstyrene ranged between 2.76 and 9.19 μg/kg, benzaldehyde ranged between 0.003 and 0.011 μg/kg, octanal ranged between 0.071 and 0.236 μg/kg, 1-hexanol and decanal ranged between 0.001 and 0.002 μg/kg.

##### Volatile Compounds Quantification

Lipid oxidation is a complex process where unsaturated fatty acids react with molecular oxygen via a free radical mechanism or in a photosensitised oxidation process. The principal source of off-flavours developed by lipid oxidation is hydroperoxides, which are unstable and readily decompose to form aliphatic aldehydes, ketones and alcohols. Many of these secondary oxidation products have undesirable odours with particularly low odour thresholds [53,54]. As it is shown in Table 3 and Table 4, the effect of studied nanoclays (T1, T2 and T3) were different depending on the polymer matrix (PHB226, PHA1005 or PHA3002). 

In general, the volatile compound quantified in greater quantity for all the samples has been the decanal as it is the volatile with the highest *M*_w_. Table 4 results suggest that there is a clear tendency which indicates that the content of the volatiles is related to the molecular weight, as the higher the *M*_w_ is, the more difficult the volatiles are to extract from the polymer and to clean during purification stages. Therefore, we can observe them in higher amounts. For the case of α-methylstyrene and benzaldehyde, these are more complex volatiles containing aromatic groups which may be also less accessible to solvents to be extracted and cleaned. Hence, they can be found also in higher quantities than simpler alcohols.

According to García-Quiles et al. [30], the addition of nanoclays may modify the structure of the polymers studied affecting the release of the studied undesirable volatile compounds. Regarding the control matrix polymers, PHB 226 initially shows lower contents of heptanal, α-methylstyrene, octanal and 1-hexanol while its content in decanal was the highest respect to the other two PHA matrices. However, benzaldehyde was initially quantified in similar amounts in all control matrices.

Concerning the effect of the addition of T1, T2 and T3, each studied volatile showed a different behaviour depending on the matrix and organoclay used (Figure 5 and Table 5).

Regarding 1-hexanol, PHB 3002 showed the lowest content in contrast to PHA1005 and PHB226. In this sample, no significant enhancement was found between the three nanoclays, although T1 at least maintained the level of release, while T3 and T2 showed non-desirable and remarkable increasing peaks of 200% and 515% respectively. For PHA 1005 all the nanoclays reduced the release of this compound, but it seems that T3 clay was much more effective reducing it in a 72%.

Heptanal, together with Decanal, seemed to be the most difficult volatile to be adsorbed by any of the organoclays. In this sense, T1 seemed to be more efficient maintaining heptanal release for PHA1005 and PHA3002 and reducing a 20% the release PHB226. In contrast, it seems that the addition of T3 could modify the structure of PHA1005 and PHA3002 polymers increasing the heptanal release in more than 400% while reducing 13% for PHB226.

Concerning octanal behaviour, the initial concentration was lower in PHB 226 but for this matrix, it seems that all organoclays provoke increases in the release of this volatile instead of retaining them. On the contrary, T3 prevents the release of octanal in PHA 1005 decreasing its release in a remarkable 92%. However, for PHA 3002, it seems that T2 suits better as it hinders the release of this volatile compound more efficiently than T3, 66%.

PHB 226 showed the highest decanal initial content. T2 was the most effective clay reducing the release of this compound in a 43%. T2 nanoclay was also the most efficient for the PHA 1005 matrix although there was no decrease for any of the three organoclays. The same tendency was found for PHA 3002, being in this case T3 the nanoclay increasing the release to a lesser extent.

Regarding α-methylstyrene, the lowest content was determined in PHB 226 and PHA1005 at equal rates by decreasing its release in 75%. For this compound and in particular for PHA 3002_T3 the developed formulation seems to be especially effective since it was not possible to detect the presence of this volatile compound therein. 

Finally, Benzaldehyde was initially present in a similar quantity in all formulations. In this case, T2 could be more effective in PHB 226 and PHA 1005 matrices whereas for PHA 3002 no significant effect was observed. 

## 4. Conclusions

The incorporation of organoclays to PHAs greatly affects the dispersion and integration of the nanoclay in the polymer matrix due to surface modification. SEM micrographs confirm how important is adequate the surface polarity of nanoclays for a particular matrix, having found largest differences between T2 which showed a lack of surface adhesion and voids formation and T3 apparently exhibiting better surface interaction and delamination. There is also a clear difference between micrographs containing T1 and T2, especially for PHB226, where the aminosilane modification clearly accentuates affinity for the PHAs matrices. Moreover, T1 has been demonstrated to be the unique nanoclay really enhancing the thermal stability of the composites developed according to TGA results while T2 maintains the original behaviour and T3 induces a remarkable decrease on T_onset_ and T_max_.

In addition, TGA analysis has allowed us to identify thermal decomposition peaks for our composites and understand the degradation phases of matrices and the decomposition and volatilization of the organic modifiers employed onto nanoclays. However, the release of volatile compounds coming from matrix aldehydes and ketones at thermal rates between 100 and 220 °C cannot be identified with this technique and therefore the determination of this off-flavour compounds has been optimised by applying headspace analysis. The obtained results from the experimental design demonstrate the suitability of the HS-SPME technique followed by GC-MS to be used in biopolymer analysis to identify and monitor the release of volatile compounds in PHAs matrices. 

It has been demonstrated that the reached effect depends on the volatile nature and the affinity of the organoclay with each matrix. On the one hand, for PHB226, T1 seems more suitable for low *M*_w_ volatiles whileT2 could be more effective to reduce the released volatiles with higher *M*_w_, which indeed are present in a major proportion. On the other hand, for PHA 1005 compounds, T2 seems to be the most effective to reduce heptanal and decanal and also α-Methylstyrene and benzaldehyde. However, T3 was of great relevance in the reduction of the octanal and 1-hexanol compounds. Finally, for PHA 3002 formulations T1 seems the most effective to reduce 1-hexanol and heptanal (lowest *M*_w_), while T2 would tackle best octanal and benzaldehyde (medium *M*_w_) and T3 would be the most effective for decanal and α-Methylstyrene (highest *M*_w_).

As a general outcome from this commercial PHA off-flavour analysis, if a plastic converter would not desire to tackle a particular subgroup of volatiles, but try to comprise as much of them as possible, T2 might result in the most versatile candidate. However, the compromise with other properties such as mechanical or barrier properties for example for food or cosmetics packaging should also be taken under consideration.

## Figures and Tables

**Figure 1 polymers-11-00945-f001:**
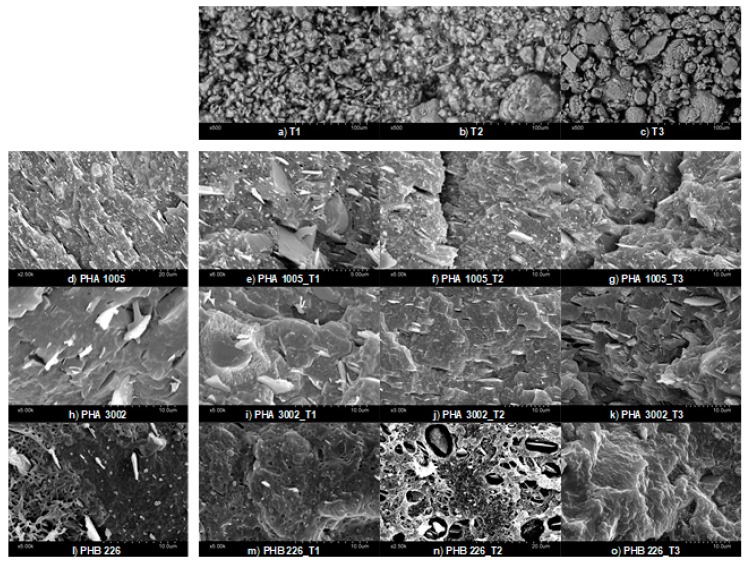
SEM micrographs of nanoclays (**a**–**c**); neat matrices (**d**,**k**,**l**) and composites: PHA1005 composites (**e**–**g**), PHA3002 composites (**i**–**k**); PHB226 composites (**m**–**o**).

**Figure 2 polymers-11-00945-f002:**
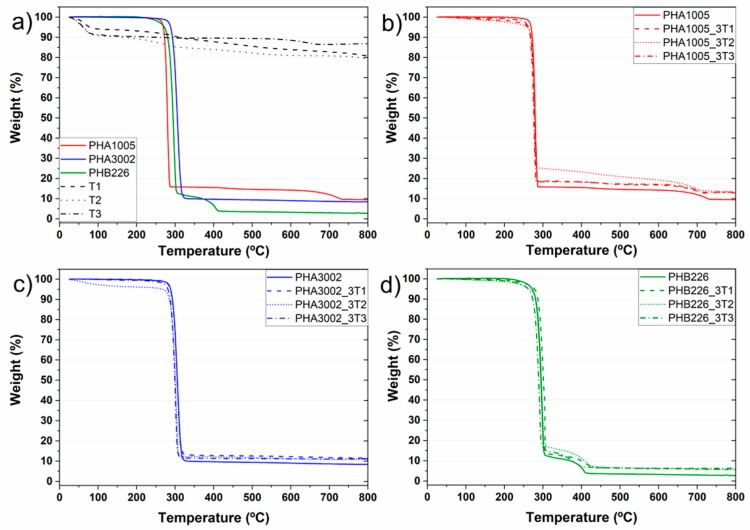
TGA curves: (**a**) neat matrices and nanoclays; (**b**) PHA1005 composites; (**c**) PHA3002 composites; (**d**) PHB226 composites.

**Figure 3 polymers-11-00945-f003:**
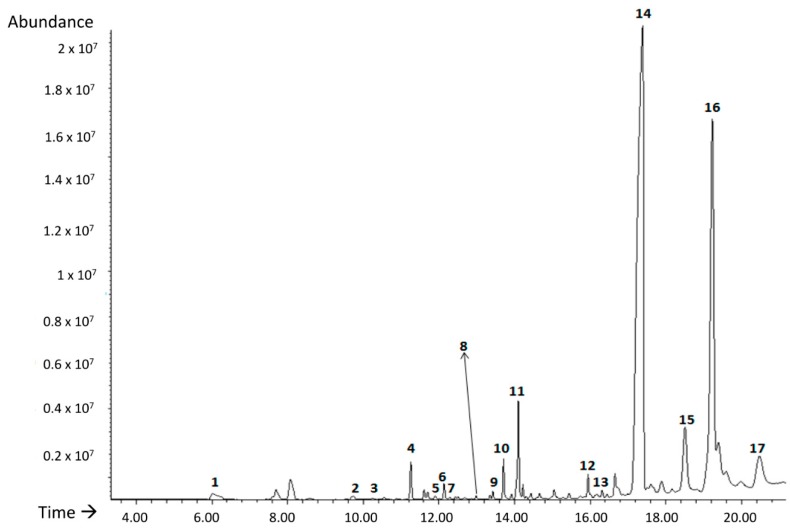
TIC chromatogram obtained for PHB 226 sample under run 3 of the BBD. ** Time refers to s*.

**Figure 4 polymers-11-00945-f004:**
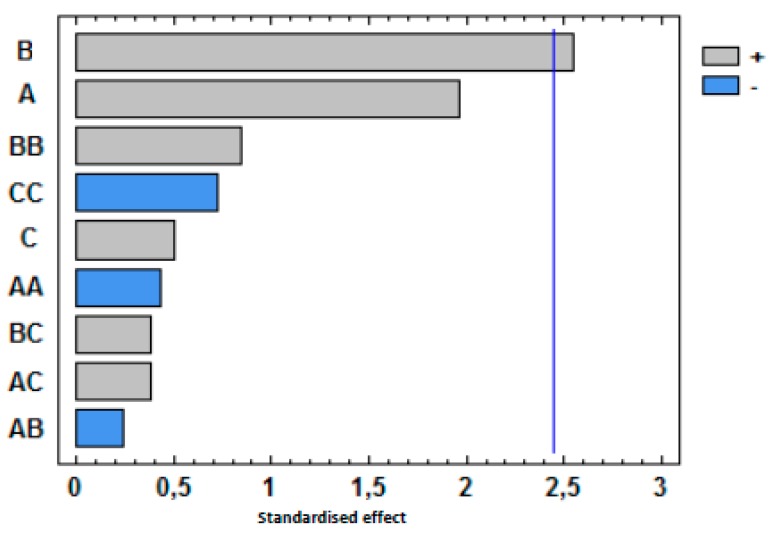
Pareto charts of factors and interactions obtained from the BBD for the sum of the volatile compounds. The vertical line indicates the statistical significance at 5% of the effects.

**Figure 5 polymers-11-00945-f005:**
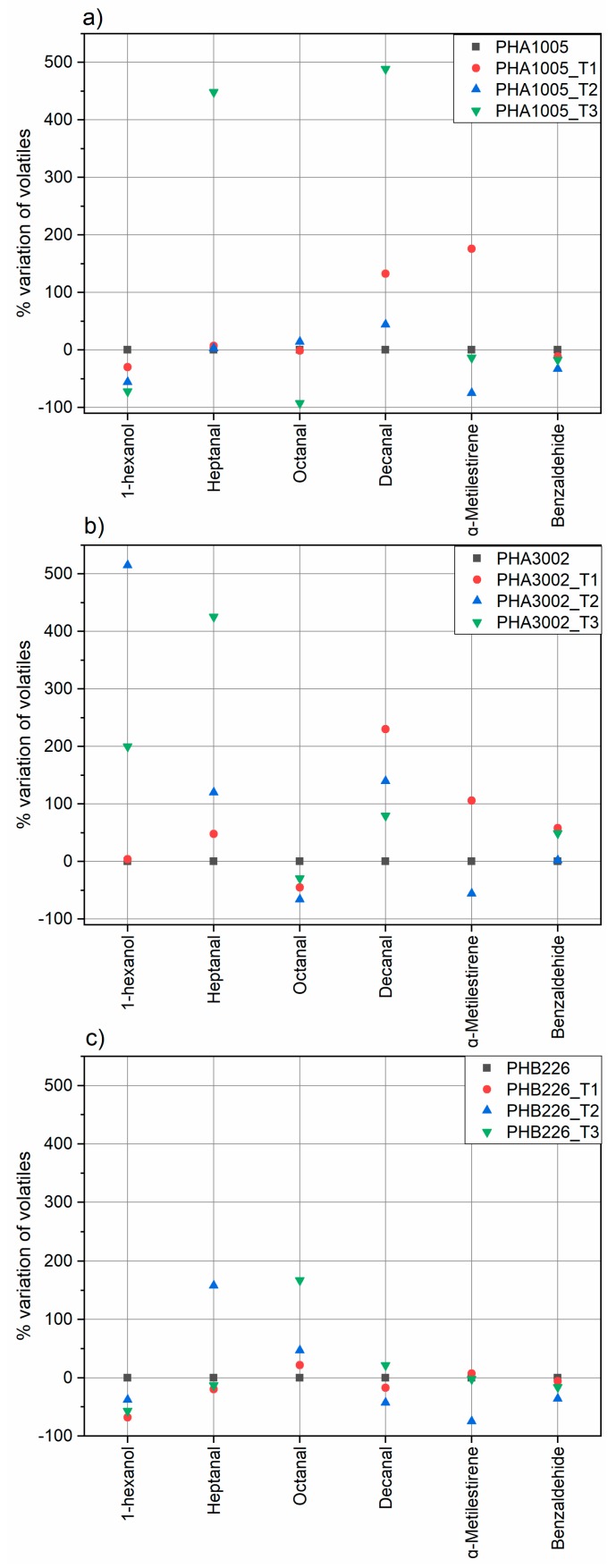
Percentage variation of volatile substances in the samples of study: (**a**) PHA1005, (**b**) PHA3002, (**c**) PHB226.

**Table 1 polymers-11-00945-t001:** Summary of material formulations. Table reproduced with permission of García-Quiles et al. [30].

Material Formulation	Commercial Matrix Used	Nature of the PHA	Type of Reinforcement (3 wt %)
PHA1005	PHA1005 (Metabolix)	P3HB-*co*-P4HB	(3HB-*co*-17 mol % 4HB) & Talc
PHA1005_T1	PHA1005 (Metabolix)	P3HB-*co*-P4HB	T1: Aminosilane Sepiolite
PHA1005_T2	PHA1005 (Metabolix)	P3HB-*co*-P4HB	T2: Natural Sepiolite
PHA1005_T3	PHA1005 (Metabolix)	P3HB-*co*-P4HB	T3: Sodium Montmorillonite-quaternary ammonium salt
PHA3002	PHA3002 (Metabolix)	P3HB-*co*-P4HB	(3HB-*co*-23.5 mol % 4HB) & Talc
PHA3002_T1	PHA3002 (Metabolix)	P3HB-*co*-P4HB	T1: Aminosilane Sepiolite
PHA3002_T2	PHA3002 (Metabolix)	P3HB-*co*-P4HB	T2: Natural Sepiolite
PHA3002_T3	PHA3002 (Metabolix)	P3HB-*co*-P4HB	T3: Sodium Montmorillonite-quaternary ammonium salt
PHB226	PHB226 (Biomer)	P3HB	Traces of PBA and plasticizer found, & Talc
PHB226_T1	PHB226 (Biomer)	P3HB	T1: Aminosilane Sepiolite
PHB226_T2	PHB226 (Biomer)	P3HB	T2: Natural Sepiolite
PHB226_T3	PHB226 (Biomer)	P3HB	T3: Sodium Montmorillonite-quaternary ammonium salt

**Table 2 polymers-11-00945-t002:** Box-Benkhen experimental design proposed for the headspace solid-phase microextraction (HS-SPME) optimization procedure.

Run	Temperature (°C)	Time (min)	NaCl (1 M)
1	70	37.5	0.5
2	50	60	0.5
3	90	60	0.5
4	70	60	0
5	90	15	0.5
6	70	15	1
7	70	37.5	0.5
8	90	37.5	0
9	90	37.5	1
10	70	15	0
11	50	37.5	1
12	70	37.5	0.5
13	50	37.5	0
14	70	60	1
15	50	15	0.5
16	70	37.5	0.5

**Table 3 polymers-11-00945-t003:** Thermal results obtained by TGA. T_onset_ %, T_max_ %, and T_50_ % represent the temperature of the initial degradation, the maximum degradation rate of decomposition, and 50 wt % loss of the samples, respectively. FR represents the final residue and DTG first derivative.

Materials	T_onset_ [°C]		T_max_ [°C]		T_50_ wt % [°C]	FR [%]
PHA1005	268.13		278.33		279.62	9.45
PHA1005_T1	267.94		275.51		278.89	12.93
PHA1005_T2	265.75		278.33		278.52	13.56
PHA1005_T3	263.20		278.33		277.25	13.27
PHA3002	290.01		307.67		305.34	8.39
PHA3002_T1	287.46		307.67		305.61	11.71
PHA3002_T2	283.63		302.33		300.23	10.93
PHA3002_T3	284.73		299.67		298.95	11.26
PHB226	275.97	386.87	297.00	403.70	293.84	2.73
PHB226_T1	283.63	384.31	299.82	411.67	301.50	5.61
PHB226_T2	278.52	383.03	297.00	407.30	296.40	5.83
PHB226_T3	272.14	347.29	291.49	382.86	287.46	6.37

**Table 4 polymers-11-00945-t004:** Volatile compounds content for the PHB 226 PHA 1005 and PHA 3002 controls and the T1, T2 and T3 formulations, expressed as the mean ± SD (n = 3).

Volatile Compound												
Sample Material	1-Hexanol		Heptanal		Octanal		Decanal		α-Methylstyrene		Benzaldehyde	
	Average	SD	Average	SD	Average	SD	Average	SD	Average	SD	Average	SD
	(μg/g sample)		(μg/g sample)		(μg/g sample)		(μg/g sample)		(μg/g sample)		(μg/g sample)	
PHA 1005	18.8	3.8	14.8	6.8	68.6	14.9	900	500	102.8	35.4	6.4	0.3
PHA 1005_T1	13.2	1.1	15.8	1.5	68.1	7.1	2100	500	283.6	12.2	5.7	1.2
PHA 1005_T2	8.2	1.3	15.3	3.2	78.1	13.9	1300	200	25.7	10.0	4.3	0.3
PHA 1005_T3	5.3	0.7	81.2	29.3	5.3	0.7	5300	700	89.6	33.5	5.3	0.7
PHA 3002	2.7	1.0	18.8	10.0	75.4	18.8	1000	300	135.4	28.4	4.5	1.5
PHA 3002_T1	2.8	1.0	27.9	1.7	41.2	6.5	3300	500	279.5	17.6	7.1	0.3
PHA 3002_T2	16.6	2.9	41.4	16.3	25.3	7.4	2400	400	59.3	11.6	4.6	0.7
PHA 3002_T3	8.1	2.7	98.8	11.1	53.6	16.8	1800	400	nd	nd	6.7	1.1
PHB 226	3.7	0.9	4.0	0.6	26.7	0.5	2300	100	40.4	4.6	7.7	0.7
PHB 226_T1	1.2	0.9	3.2	1.1	32.6	6.0	1900	200	43.4	7.1	7.2	2.6
PHB 226_T2	2.3	0.6	10.3	0.9	39.3	7.7	1300	300	10.3	2.3	4.9	0.5
PHB 226_T3	1.6	0.6	3.5	7.6	71.3	17.9	2800	700	39.6	17.3	6.5	2.7

nd = no data.

**Table 5 polymers-11-00945-t005:** Summary of the organoclay that works best to reduce the release of the volatile compounds studied for each polymer matrix.

Volatile	PHA 1005		PHA 3002		PHB 226
1-Hexanol	T3		T1		T1
Heptanal	T2		T1		T1
Octanal	T3		T2		T1
Decanal	T2		T3		T2
α-Methylstyrene	T2		T3		T2
Benzaldehyde	T2		T2		T2

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
