# Peer review of "Reducing off-Flavour in Commercially Available Polyhydroxyalkanoate Materials by Autooxidation through Compounding with Organoclays"

_polymers, 2019, doi:10.3390/polym11060945_

Round 1
Reviewer 1 Report
The manuscript is very interesting and well written. It refers to very popular actual issue of the displeasing fragrance referring to polyhydroxyalkanoates. The authors propose bionanocomposites manufacture based on polyhydroxyalkanoates with organomodified nanoclays. these bionanocomposites have capability to capture volatile compounds responsible for displeasing aroma. In order to check the adsorbance properties new composites, authors used the known research technique i.e. headspace solid-phase microextraction (HS-SPME) which was not earlier used in this type tests. The obtained results are interesting and promising taking into account the practical significance of the considered issue. The used research methods are suitable and fully illustrate the problem under consideration.
I think that the manuscript should be published after minor correction.
The characterization of the tested polyhydroxyalkanoates should be described in the manuscript.
Authors mentioned that mechanic properties of the bionanocomposites are out of scope of this paper. I think that authors should indicate suitable reference with description of these properties.
In figure 3, there are some peaks which were not identified, why?
The table 4 should be rearranged and the sample materials should be grouped according to the type of polyhydroxyalkanoate.
English in manuscript should be checked, small mistakes are present, e.g. page 1, line 25 organomidified.
Author Response
REVIEWER #1
The manuscript is very interesting and well written. It refers to very popular actual issue of the displeasing fragrance referring to polyhydroxyalkanoates. The authors propose bionanocomposites manufacture based on polyhydroxyalkanoates with organomodified nanoclays. these bionanocomposites have capability to capture volatile compounds responsible for displeasing aroma. In order to check the adsorbance properties new composites, authors used the known research technique i.e. headspace solid-phase microextraction (HS-SPME) which was not earlier used in this type tests. The obtained results are interesting and promising taking into account the practical significance of the considered issue. The used research methods are suitable and fully illustrate the problem under consideration.
I think that the manuscript should be published after minor correction.
The characterization of the tested polyhydroxyalkanoates should be described in the manuscript.
Authors mentioned that mechanic properties of the bionanocomposites are out of scope of this paper. I think that authors should indicate suitable reference with description of these properties.
In figure 3, there are some peaks which were not identified, why?
The table 4 should be rearranged and the sample materials should be grouped according to the type of polyhydroxyalkanoate.
English in manuscript should be checked, small mistakes are present, e.g. page 1, line 25 organomidified.
REPLY to reviewer #1
We really thank reviewer 1 for his/her positive comments. We answer below to his/her suggestion point by point:
Regarding the characterization of PHA and the mechanical properties suggestion, we have included a sentence indicating where these can be found:
“A complete characterization covering the structural, thermal and mechanical behaviour of the formulations developed, understanding the compatibility mechanisms between the different organoclays and the matrices can be found in a previous work carried out by authors and published at García-Quiles et al.).[38]”
As a complete characterization of the composites developed has already been recently published in Polymers:
García-Quiles, L., Fernández, A., Castell., P. Sustainable Materials with Enhanced Mechanical Properties Based on Industrial Polyhydroxyalkanoates Reinforced with Organomodified Sepiolite and Montmorillonite, Polymers 2019, 11(4), 696. doi:10.3390/polym11040696
· Regarding Figure 3, Peaks were identified and when standards were not available, volatile compounds having ≥80% similarity with Wiley library were tentatively identified using GC-MS spectra only. This information has been added to the text lines: 183-185, to clarify it for the reader.
· Concerning table 4 we have followed the reviewer suggestion and the table lines have been rearranged grouped according to the type of polyhydroxyalkanoate.
· Finally, we have carried out a careful checking of the English to erase some typos and enhance the overall quality and understanding of the document.
Reviewer 2 Report
Three different organoclays (1 natural, 2 modiefied) from the company TOLSA where used as post additives in commercially available PHA materials, with the goal to reduce PHA specific odor. Each PHA material was compounded at 140 – 160 °C with each 3%wt of the additives. The effect on the thermostability was investigated by TGA analyses, SEM imaging was used to monitor the change in surface area. Head-space GC-MS was used to quantify 6 volatiles (mainly aldehydes) in the PHA materials and the different created compounds, to investigate the effect of the added adsorbents. The sample preparation for the GC-MS was optimized prior sample analyses.
The usage of adsorbents to reduce PHA specific odor is an interesting investigation. However, adsorbents should be used in the initial formulation step during PHA processing to avoid a second compounding process. First, this is not economic, second this second compounding process will most likely interfere with the material properties. This facts should be discussed in the manuscript.
However the presented findings are interesting and could be useful for people working on optimization of PHA additives. Nevertheless the manuscript has essential weakness in the presentation, description and discussion of the data. The English should be carefully checked and the phrase “on the other hand” is way too much used in the manuscript. PHA grades should be changed to PHA materials
Major comments
1. Title should be changed to: Reducing off-flavor in commercially available Polyhydroxyalkanoate materials by autooxidation through compounding with organoclays
2. Separate Result and Discussion section. In the current version the manuscript is hard to read.
3. Table 1
a. Include a column with PHA supplier
b. When was the material purchased? Metabolix does not exist since two years
c. How was the concentration of 3wt% chosen?
d. What does 17% P4HB/P3HB mean? P(3HB-co-17mol%4HB)?
4. Compounding
a. Describe more in detail give quantities of used PHA materials
b. For the control the PHA material was treated the same?
c. Either way include mechanical property data or take out the last two sentences (line 142 – 144)
5. Table 3
a. Control data are from the PHA materials as purchased or after compounding treatment with no adsorbents? Either way add the missing data
b. Why is it _3T1 and not T1 as in the other descriptions. Is there a difference?
6. Figure 2
a. See comment 5a
7. Figure 5
a. x-axis is in µg/g but values from decanal are in mg/g
b. order should be changedPHA, PHA-T1, PHA-T2, PHA-T3
c. Line 402 to 431 check all the statements to increase or decrease of the volatile substances in the compounds there are many wrong statements e.g.
i. Line 410 statement about T 1 not correct
8. Table 5
a. Take out does not give any useful information
9. New figure
a. Create a new figure with % increase or decrease of the volatile substances for the different adsorbents
10. Disussion
a. After separation of the discussion from the result section include the comments made above.
b. discuss why volatile are in or decreasing
11. Conclusion Line 456 heptanal AND octanal
12. References
a. Line 48: Replace Reference 3 with: reference
REHM, B. H. A. (2003) ‘Polyester synthases: natural catalysts for plastics’, Biochemical Journal, 376(1), pp. 15–33. doi: 10.1042/bj20031254.
b. Line 52ff: A sentences about potential carbon sources for PHA production is needed in the introduction, since the used carbon source might have an impact on the odor of the final PHA. (Brigham and Riedel, 2018) gives a recent overview of the PHA production from food wastes.
Brigham, C. J. and Riedel, S. L. (2018) ‘The Potential of Polyhydroxyalkanoate Production from Food Wastes’, Applied Food Biotechnology, 6(1), pp. 7–18.
c. Line 60: add a reference mentioning also biological PHA recovery: (Ong et al., 2017)
Ong, S. Y. et al. (2017) ‘An integrative study on biologically recovered polyhydroxyalkanoates (PHAs) and simultaneous assessment of gut microbiome in yellow mealworm’, Journal of Biotechnology. Elsevier, 265(June 2017), pp. 31–39. doi: 10.1016/j.jbiotec.2017.10.017.
d. Line 62: With the statement “currently under investigation” you can not cite literature from 2011 and 2008
Minor comments
Line 32: take out (T2)
Line 34 – 36: reduce key words
Line 50: hydroxybutyrate-co-hydroxyvalerate
Line 54: PHA are used as carbon and energy storage should be added
Line 157: 1 ml à 1 mL
Line 163: 10 minutes à 10 min.
Table 4: sort after type and chain length
Line 399: table 3 à Table 3
Layout
scl-PHA à scl-PHA
use nonbreaking space after between value and unit
Author Response
REVIEWER #2
We deeply thank the editor for the detailed reading carried out looking for an excellent publication.
Three different organoclays (1 natural, 2 modified) from the company TOLSA where used as post additives in commercially available PHA materials, with the goal to reduce PHA specific odor. Each PHA material was compounded at 140 – 160 °C with each 3%wt of the additives. The effect on the thermostability was investigated by TGA analyses, SEM imaging was used to monitor the change in surface area. Head-space GC-MS was used to quantify 6 volatiles (mainly aldehydes) in the PHA materials and the different created compounds, to investigate the effect of the added adsorbents. The sample preparation for the GC-MS was optimized prior sample analyses.
The usage of adsorbents to reduce PHA specific odor is an interesting investigation. However, adsorbents should be used in the initial formulation step during PHA processing to avoid a second compounding process. First, this is not economic, second this second compounding process will most likely interfere with the material properties. This facts should be discussed in the manuscript.
One extrusion compounding process is necessary due to the use of commercial PHAs (not synthetized in a lab). These materials present certain limitations being the odour a major one which is very difficult to avoid. The authors are aware about the fact that these adsorbents ideally should be introduced by the materials manufacturers. Authors present a methodology and the aim with the work is to show first that including organoclays may solve the problem. Of course, in a future application it will only include one stage/process at manufacturer’s stage, while in the meantime it can be a solution to be applied by converters. The justification for applying this methodology is explained in the text, lines: 62-78. In addition, we have added a sentence in the manuscript text to stress that this problem is not currently solved by PHA manufacturers. Line 75-76.
However the presented findings are interesting and could be useful for people working on optimization of PHA additives. Nevertheless the manuscript has essential weakness in the presentation, description and discussion of the data. The English should be carefully checked and the phrase “on the other hand” is way too much used in the manuscript. PHA grades should be changed to PHA materials
Thanks for the comment, your suggestions regarding English have been modified, and the text has been checked.
Major comments
1. Title should be changed to: Reducing off-flavor in commercially available Polyhydroxyalkanoate materials by autooxidation through compounding with organoclays
Authors deeply thank the reviewer for this suggestion. The title has been changed accordingly.
2. Separate Result and Discussion section. In the current version the manuscript is hard to read.
Authors have followed the comments of the three reviewers to enhance the overall manuscript, order some parts and adapt text in order to ease the understanding and reading of the article, and we really thank the reviewers for this effort. Authors have also followed the format/template indicated by Polymers magazine, in which Results and Discussion sections are not split. Therefore, authors are including all modifications suggested by reviewers in text, graphs and tables, but we do not consider necessary separating this section in two.
3. Table 1
a. Include a column with PHA supplier
The supplier has been included between brackets in the “commercial matrix used” column. It is also indicated just after table one in lines 122- 124.
b. When was the material purchased? Metabolix does not exist since two years
The material was purchased about 2,5 years ago. The development of the methodology, method, analysis etc was carried out then, although we are publishing the results now.
c. How was the concentration of 3wt% chosen?
Authors have been working with these types of organoclays for different purposes for several years with vary materials. In particular we have worked with this family of Sepiolites previously with other biopolymers such as PLA and we have published some results; e.g.:
Peinado, V.; García, L.; Fernández, A.; Castell, P. Novel lightweight foamed poly(lactic acid) reinforced with different loadings of functionalised Sepiolite, Composites Science and Technology 2014, 101, 17–23. DOI: 10.1016/j.compscitech.2014.06.025
Peinado, V., Castell, P., García, L., & Fernández, Á. Effect of Extrusion on the Mechanical and Rheological Properties of a Reinforced Poly(Lactic Acid): Reprocessing and Recycling of Biobased Materials. Materials 2015, 8(10), 7106–7117. doi:10.3390/ma8105360
The whole work carried out pretends to solve this off-flavour problem why maintaining or even enhancing thermal and mechanical properties. Our experience have led to a selection between 2wt% to 4wt%, no less no more (due to agglomerates). We have also corroborated this data with literature to understand if we were in coherence with other authors. Some of them are cited as references in this work; e.g.:
Wang, S.; Song, C.; Chen, G.; Guo, T.; Liu, J.; Zhang, B.; Takeuchi, S. Characteristics and biodegradation properties of poly(3-hydroxybutyrate-co-3-hydroxyvalerate)/organophilic montmorillonite (PHBV/OMMT) nanocomposite, Polymer Degradation and Stability 2005, 87, 69-76.
González-Ausejo, J., Gámez-Pérez, J., Balart, R., Lagarón, J. M., Cabedo, L. Effect of the addition of sepiolite on the morphology and properties of melt compounded PHBV/PLA blends, Polymer Composites 2017, doi:10.1002/pc.24538
d. What does 17% P4HB/P3HB mean? P(3HB-co-17mol%4HB)?
Yes, that is exactly the meaning, we re-write it as reviewer suggests modifying table 1.
4. Compounding
a. Describe more in detail give quantities of used PHA materials
Ok, it has been included: A total amount of 3 to 5 kg per material was produced. Lines 143.
b. For the control the PHA material was treated the same?
Yes, it was treated in the same way.
c. Either way include mechanical property data or take out the last two sentences (line 142 – 144)
Regarding the characterization of PHA and the mechanical properties suggestion, we have included a sentence indicating where these can be found (lines 147-150):
“A complete characterization covering the structural, thermal and mechanical behaviour of the formulations developed, understanding the compatibility mechanisms between the different organoclays and the matrices can be found in a previous work carried out by authors and published at García-Quiles et al.).[38]”
As a complete characterization of the composites developed has already been recently published in Polymers:
García-Quiles, L., Fernández, A., Castell., P. Sustainable Materials with Enhanced Mechanical Properties Based on Industrial Polyhydroxyalkanoates Reinforced with Organomodified Sepiolite and Montmorillonite, Polymers 2019, 11(4), 696. doi:10.3390/polym11040696
5. Table 3
a. Control data are from the PHA materials as purchased or after compounding treatment with no adsorbents? Either way add the missing data
All materials have been processed under the same conditions as described in the experimental section.
b. Why is it _3T1 and not T1 as in the other descriptions. Is there a difference?
This is a Typo, many thanks (it is the same).
6. Figure 2
a. See comment 5a
All materials have been processed under the same conditions as described in the experimental section.
7. Figure 5
Figure 5 has been substituted by a new Figure, as suggested in point 9.
a. x-axis is in µg/g but values from decanal are in mg/g
Because Decanal has been detected at mg/g level (much higher release than the others). For this reason the suggested new figure showing percentages of increase and decrease is rather useful and has been changed.
In Table 4 (line 401) we have put Decanal in µg/g, to avoid mixing units.
b. order should be changedPHA, PHA-T1, PHA-T2, PHA-T3
Figure deleted
c. Line 402 to 431 check all the statements to increase or decrease of the volatile substances in the compounds there are many wrong statements e.g.
i. Line 410 statement about T 1 not correct
The statements have been checked and the text modified and adapted to the table 4 and new figure 5 results. The whole text has been carefully reviewed and statements have been fine-tuned. Lines 421-453.
8. Table 5
a. Take out does not give any useful information
Authors understand that this table is the summary or conclusion of the best organoclay to be used to tackle the studied volatiles. We understand that there may be readers of Polymers with a different background and, in case the reader is not too familiar with the HS-SPME-GC-MS technology and output, it can help to understand the message of the research carried out. In addition, the table has been checked and finally only one organoclay per volatile and matrix has been indicated (the best candidate) eliminating some multiple-choice we gave in a beginning.
9. New figure
a. Create a new figure with % increase or decrease of the volatile substances for the different adsorbents
The figure has been created and will substitute current Figure 5, as authors consider that data is shown in Table 4, and the new figure suggested will better help the reader to follow the increase and decrease of volatile substances rather than the previous Figure. We deeply thank the reviewer for this very useful suggestion.
10. Disussion
a. After separation of the discussion from the result section include the comments made above.
Authors have followed the comments of the three reviewers to enhance the overall manuscript, order some parts and adapt text in order to ease the understanding and reading of the article, and we really thank the reviewers for this effort. Authors have also followed the format/template indicated by Polymers magazine, in which Results and Discussion sections are not split. Therefore, authors are including all modifications suggested by reviewers in text, graphs and tables, but we do not consider necessary separating this section in two.
c. discuss why volatile are in or decreasing
The new figure 5 and the re-order of table 4 has served to clarify the tendencies (for increasing and decreasing). Thank you for such input. It can be observed that the tendency on the content of the volatiles is related to the molecular weight, as the higher the Mw is, more difficult the volatiles are to be extracted from the polymer and be cleaned during purification stages. Therefore, we can see them in higher amounts (being Decanal found even in mg/g). For the case of ɑ-Methylstyrene and Benzaldehyde, these are more complex volatiles containing aromatic groups which may be also less accessible to solvents to be extracted and cleaned. Hence, they can be found also in higher quantities. Partners have included a paragraph explaining this tendency lines 415-420.
11. Conclusion Line 456 heptanal AND octanal
Yes, included.
12. References
a. Line 48: Replace Reference 3 with: reference
REHM, B. H. A. (2003) ‘Polyester synthases: natural catalysts for plastics’, Biochemical Journal, 376(1), pp. 15–33. doi: 10.1042/bj20031254.
Authors have included the reference suggested
b. Line 52ff: A sentences about potential carbon sources for PHA production is needed in the introduction, since the used carbon source might have an impact on the odor of the final PHA. (Brigham and Riedel, 2018) gives a recent overview of the PHA production from food wastes.
Brigham, C. J. and Riedel, S. L. (2018) ‘The Potential of Polyhydroxyalkanoate Production from Food Wastes’, Applied Food Biotechnology, 6(1), pp. 7–18.
The suggested reference has been included. In addition, a sentence has been added to mention the carbon source as suggested. Lines 60-61:
Recent works have studied the effect that carbon sources may have on the final odours released by PHA monomers [11].
c. Line 60: add a reference mentioning also biological PHA recovery: (Ong et al., 2017)
Ong, S. Y. et al. (2017) ‘An integrative study on biologically recovered polyhydroxyalkanoates (PHAs) and simultaneous assessment of gut microbiome in yellow mealworm’, Journal of Biotechnology. Elsevier, 265(June 2017), pp. 31–39. doi: 10.1016/j.jbiotec.2017.10.017.
We have included the suggested reference (10)
d. Line 62: With the statement “currently under investigation” you can not cite literature from 2011 and 2008
Authors have changed it for “have been investigated”.
Minor comments
Line 32: take out (T2)à Ok, removed
Line 34 – 36: reduce key words à We are pending on editor’s final decision, thank you.
Line 50: hydroxybutyrate-co-hydroxyvalerate à changed
Line 54: PHA are used as carbon and energy storage should be added à added
Line 157: 1 ml à 1 mL à changed
Line 163: 10 minutes à 10 min. à changed
Table 4: sort after type and chain length à Modified as suggested
Line 399: table 3 à Table 3 à changed
Layout
scl-PHA à scl-PHA à changed
use nonbreaking space after between value and unit à revised and changed
Reviewer 3 Report
This study describes the preparation of commercial PHA filled with nanoclys to entrap flavors (determined by particular voaltiles as defined by the auhtos). The work is original, it reports interesting results and is well prepared. I recommend its publication after addressing properly the following points:
Introduction
Please note in the sentence “These features give rise to diverse PHA combinations with tailored molecular weights and melting points providing broad properties, such as…” some of the materials described are not polymers but monomers to build PHAs. Then amend the sentence.
Information included in the Introduction from line 77-95 describes a technique well known for gas analysis. In my opinion, this information is not necessary. I suggest to remove it and include previous works using polymer composites mineral fillers with adsorption capacity.
Lines 96-97 correspond to the materials section.
Experimental
The information related to the composition of the commercial PHA should be supported by any reference or analysis since this is not disclosed by the manufacturers.
I suggest to include the temperature profile of each formulation in Table 1 and remove it from section 2.2. Please also include the processing conditions during injection molding to shape the composites.
Add full details applied during TGA analysis.
Results
3.1. The usefulness of having the DTG values in the table seems unclear
3.2. Since the absorption capacity of the fillers is based on a porosity phenomenon images at higher magnification are needed. I recommend the use of TEM in case this cannot be achieved by SEM. Please also check the organization and codification of samples in Figure 2. Also describe how the surface fractures were obtained. I recommend to start the results section with morphology and the thermal and active properties.
3.3. In order to better ascertain the effect of each filler on the absorption capacity I suggest in Table 4 in Figure 5 to group the materials according to the type of PHA matrix. Authors should better state and describe the reasons by which each PHA contain different amounts of volatiles. The sentence in line 411 “In contrast, it seems that the addition of T3 could modify the structure of PHA polymers increasing the heptanal release.” Should be improved. It is difficult to consider that the presence of a filler can modify the structure of a polymer. This effect can be related to a chain scission phenomenon during extrusion or hydrolytic degradation. Authors should better explain this part and support the results with references. I also suggest to remove Table 5 and include a discussion section describing the effect of each clay on the specific absorption of any chemical. This process should be ascribed to a physical process (porosity) but the fact that it shows a significant variation for each volatile indicates that the chemistry on the clay surfaces is also playing a major role in this effect. Finally, please use the statistical analysis to determine if the changes observed on the absorption are significant.
Conclusions
Propose real examples and benefits of the application of the resultant composites, for instance food packaging.
Author Response
REVIEWER #3
This study describes the preparation of commercial PHA filled with nanoclays to entrap flavors (determined by particular volatiles as defined by the authors). The work is original, it reports interesting results and is well prepared. I recommend its publication after addressing properly the following points:
Introduction
Please note in the sentence “These features give rise to diverse PHA combinations with tailored molecular weights and melting points providing broad properties, such as…” some of the materials described are not polymers but monomers to build PHAs. Then amend the sentence.
Thanks for the suggestion, the sentence has been corrected accordingly. (Line 47-50)
Information included in the Introduction from line 77-95 describes a technique well known for gas analysis. In my opinion, this information is not necessary. I suggest to remove it and include previous works using polymer composites mineral fillers with adsorption capacity.
Thanks for the suggestion. Although HS-SPME is a well-known technology for chemists, it may be not so well known for readers with other backgrounds. In addition, it has not been previously applied in this methodology, therefore the reader may be not so familiar with it also in this area. For these reasons authors understand that the description helps to the overall understanding of the work carried out.
Lines 96-97 correspond to the materials section.
OK, the sentence has been amended according to the suggestion (now, lines 98-99)
Experimental
The information related to the composition of the commercial PHA should be supported by any reference or analysis since this is not disclosed by the manufacturers.
Thank you for the suggestion. All the information regarding characterization & composition of the commercial materials is described elsewhere (as a complete characterization of the composites developed has already been recently published in Polymers):
García-Quiles, L., Fernández, A., Castell., P. Sustainable Materials with Enhanced Mechanical Properties Based on Industrial Polyhydroxyalkanoates Reinforced with Organomodified Sepiolite and Montmorillonite, Polymers 2019, 11(4), 696. doi:10.3390/polym11040696
A small paragraph and the above reference has been added in the materials and methods section indicating where the information can be found.
I suggest to include the temperature profile of each formulation in Table 1 and remove it from section 2.2. Please also include the processing conditions during injection molding to shape the composites.
Thank you for the suggestion, However Table 1 with all the modifications suggested by reviewers seems now a bit overloaded and we would be duplicating information. Therefore authors will keep the temperature profile of each formulation described inside point 2.2. Nano-bio-composites preparation.
These profiles are:
· PHA1005 and PH3002: 150 ºC in the feeding zone up to 165 ºC at the nozzle
· PB226: 140 ºC in the feeding zone up to 160 ºC at the nozzle
Add full details applied during TGA analysis.
Further details regarding TGA analysis have been added in lines 161-162
Results
3.1. The usefulness of having the DTG values in the table seems unclear
Thank you for the suggestion. Yes, it is not almost described during the discussion, therefore we are removing that column as it may not provide useful information for the reader.
3.2. Since the absorption capacity of the fillers is based on a porosity phenomenon images at higher magnification are needed. I recommend the use of TEM in case this cannot be achieved by SEM. Please also check the organization and codification of samples in Figure 2. Also describe how the surface fractures were obtained. I recommend to start the results section with morphology and the thermal and active properties.
Thank you for the comment, unfortunately authors have not possibility in the frame of this project to run a TEM trial, and at least with SEM we can draw some conclusions (sure TEM could improve further).
Yes, there was a typo in the codification of samples in Figure 2 as it can be read as PHA1005_3T1 instead of PHA1005_T1 …and so on. We have corrected this typo.
Authors have included a sentence describing how the surface of fracture were obtained (from mechanical specimens) in lines 155-156.
Finally, authors have followed the reviewer recommendation about starting the results & discussion section with the morphology and then follow by thermal and active properties.
3.3. In order to better ascertain the effect of each filler on the absorption capacity I suggest in Table 4 in Figure 5 to group the materials according to the type of PHA matrix.
The materials have been grouped according the type of PHA in Table 4 and new figure 5 (affected by comments of other reviewers).
Authors should better state and describe the reasons by which each PHA contain different amounts of volatiles.
Regarding the reasons by which PHA contains different amounts of volatiles, this is probably due to the complex production process to obtain the PHAs as explained in the introduction, as the sub-products coming from the fermentation step are different for each PHA material. In addition, it can be observed that the tendency on the content of the volatiles is related to the molecular weight, as the higher the Mw is, more difficult the volatiles are to be extracted from the polymer and and be cleaned during purification stages. Therefore, we can see them in higher amounts (being Decanal found even in mg/g). For the case of ɑ-Methylstyrene and Benzaldehyde, these are more complex volatiles containing aromatic groups which may be also less accessible to solvents to be extracted and cleaned. Hence, they can be found also in higher quantities. Partners have included a paragraph explaining this tendency lines 415-420.
The volatiles have been identified using patterns (when standards were not available, volatile compounds having ≥80% similarity with Wiley library were tentatively identified using GC-MS spectra only) and focusing only on those with highest amount and that were considered candidates for the unpleasant rancid odour (basically aldehydes and ketones).
The novelty of the article relays in the methodology developed that permits to identify key volatiles and analyze the effect induced by the nanoclays as adsorbents, so that this methodology could be used for other materials in the future.
The sentence in line 411 “In contrast, it seems that the addition of T3 could modify the structure of PHA polymers increasing the heptanal release.” Should be improved. It is difficult to consider that the presence of a filler can modify the structure of a polymer. This effect can be related to a chain scission phenomenon during extrusion or hydrolytic degradation. Authors should better explain this part and support the results with references.
A complete characterization of the composites developed has already been recently published in Polymers:
García-Quiles, L., Fernández, A., Castell., P. Sustainable Materials with Enhanced Mechanical Properties Based on Industrial Polyhydroxyalkanoates Reinforced with Organomodified Sepiolite and Montmorillonite, Polymers 2019, 11(4), 696. doi:10.3390/polym11040696
The characterisation includes: NMR, DSC, XRD & Mechanical results. XRD shows the exfoliation of T3 (disappearing of certain key peaks) and also DSC shows modifications in the crystallization rates induces by the organoclays and the modifications are justified. Authors have included the reference in this section of the text and for more information that reader is welcome to check this other publication.
I also suggest to remove Table 5 and include a discussion section describing the effect of each clay on the specific absorption of any chemical. This process should be ascribed to a physical process (porosity) but the fact that it shows a significant variation for each volatile indicates that the chemistry on the clay surfaces is also playing a major role in this effect. Finally, please use the statistical analysis to determine if the changes observed on the absorption are significant.
Authors understand that Table 5 is the summary or conclusion of the best organoclay to be used to tackle the studied volatiles. We understand that there may be readers of Polymers with a different background and, in case the reader is not too familiar with the HS-SPME-GC-MS technology and output, it can help to understand the message of the research carried out. In addition, the table has been checked and finally only one organoclay per volatile and matrix has been indicated (the best candidate) eliminating some multiple-choice we gave in a beginning. Anyway, we will check with editor and leave this final decision for him.
Following the suggestions and recommendations of all the reviewers, authors have made major modifications in this section. In particular Figure 5 has been changed for a new figure which indicates the % increase or % decrease of each volatile substance for the different adsorbents.
Finally the results and discussion section related to the volatile compounds quantification has been checked and modified to ease the understanding of the volatiles outcome.
Authors understand that with all the suggestions made Reviewer 3 and the other 2 reviewers this section has been deeply enhanced.
Conclusions
Propose real examples and benefits of the application of the resultant composites, for instance food packaging.
Thank you for the suggestion. Authors included the potential sectors of application in the introduction (including references) lines 71-74, and also following this recommendation we have also included an example at the end of the conclusions to better close the article:
However, the compromise with other properties such us mechanical or barrier properties for example for food or cosmetics packaging should also be taken under consideration.
Round 2
Reviewer 2 Report
The authors addressed all comments well.
Reviewer 3 Report
Although the authors have given some explanations, not all of the remarks listed in review have been applied. In any case, I recommend this paper for publication.